

# Evaluation of a seed drill for barley and vetch sowing as a function of varying seeding ratios

Arzu Yazgi[1], Tuncay Gunhan[1], Behcet Kir[2], Gulcan Demiroglu Topcu[2] and Erdem Aykas[1]

[1] Agricultural Engineering and Technologies, Faculty of Agriculture, Ege University, Izmir, Turkey
[2] Field Crops, Faculty of Agriculture, Ege University, Izmir, Turkey

## ABSTRACT

**Objectives.** The objective of this study was to evaluate the performance of a seed drill for seeding barley and vetch mixtures having different ratios and to determine their impact on plant emergence, yield and feed quality of component crops.

**Methods**. The laboratory and field experiments were conducted to determine the performance of the seeder using pure barley, common vetch, and their mixtures in ratios, namely 100:0, 0:100, 75:25, 50:50, 25:75. The seed flow and distribution uniformity across the rows were determined within laboratory conditions, while other response variables were assessed in the field conditions.

**Results**. The coefficient of variation values of flow evenness for barley and vetch were found to be 1.0–5.5% and 0.3–2.1%, respectively. Seed distribution uniformity of each row unit were also determined ranging from 4.2% to 10.7% and from 0.4% to 1.4% for barley and vetch, respectively. The goodness criteria values ranged between 66.4%–86.0% for the laboratory tests while the corresponding values had a range of 78%–86% for field samples. Based on the findings in this work, the overall ranges of variation factor values were 0.43–1.28 and 0.36–0.77 for laboratory and field evaluations, respectively. Furthermore, the maximum yield was 42,620 kg ha$^{-1}$, whereas the minimum dry material ratio was 16.93% recorded for pure barley crop.

**Conclusions**. It appears that the results in this work demonstrated that the seed drill could have a great potential to be used effectively for the seeding of barley as well as vetch mixtures as a function of various ratios to enhance the overall yield of the crop.

# INTRODUCTION

In forage crop production different intercropping systems, the cultivation of two or more crops simultaneously on the same land at the same time, are used to increase the biomass yield on a per unit land area (*Khan et al., 2024*; *Iqbal, Iqbal & Abbas, 2018*; *Acar et al., 2006*). The compatibility among component crops in terms of utilizing growth resources in spatial and temporal dimensions are crucial to achieve the added advantage offered by the intercropping systems (*Abbas et al., 2021*; *Iqbal et al., 2021a*; *Bakoglu, 2004*). Also, it has been reported to reduce the competition for growth resources among component crops

Corresponding author
Arzu Yazgi, arzu.yazgi@ege.edu.tr

to avoid serious reduction in their yield and quality (*Iqbal et al., 2019*). Different biomass crops such as vetch, forage and grass pea are sown with cereals in different mixture ratios. Cereals, however, tend to dominate the leguminous crops in terms of acquiring the growth resources especially nutrients (*Abbas et al., 2021*; *Iqbal et al., 2018*). Moreover, perennial forage crops, which grow slowly in the seedling stage, can be sown with companion crops especially in the case of cereals, which are annual and rapidly grown, for weed control.

Among agronomic practices, sowing of seeds at the right depth and accuracy of desired seed spacing in the soil are crucial factors that ultimately contribute towards germination and growth of the crops (*Iqbal et al., 2021b*; *Ahmad et al., 2020*). Generally, centrifugal fertilizer spreaders such as broadcast seeders or seed drills have been used for the sowing of seed mixtures which require mixing of different seeds at different ratios following the filling process of the seed hopper, however, this method could not maintain seed spacing.

The agronomic studies on the mixture of vetch and barley conducted by previous investigations mostly concentrated on plant efficiency, feed quality and land equivalent ratio at different mixture ratios in small-scale parcels. *Soya, Avcioglu & Geren (1996)* inferred that a comparatively lesser ratio of vetch (55%) in the vetch-barley mixtures increased the yield. *Bakoglu (2004)* found the highest value of land equivalent ratio of 1.11 for the mixture of 90% vetch + 10% barley while the lowest value was 0.71 for the mixture of 60% vetch + 40% barley. In addition, the mixtures of 70% vetch + 30% barley and 80% vetch + 20% barley mixtures were suggested for obtaining a higher herbage yield.

In a study conducted by *Kumar & Durairaj (2000)*, it was found that distributor geometry was an important parameter of seed distribution uniformity. Likewise, *Uygan & Guler (2005)* found that the ideal distributor head type was T type and air velocity was 26 m s$^{-1}$. Additionally, *Bayhan et al. (2009)* examined seeding rate and variation of seed distribution evenness based on the different PTO values and found that the best value was 300 min$^{-1}$. Moreover, *Yazgi et al. (2012)* investigated the performance of the pneumatic seed drill in barley seeding and inferred that barley could be sown effectively but its performance decreased for mixed seeding of component crops.

In a study conducted by *Yazgi et al. (2017)* it was inferred that seed drill performance was determined satisfactory in terms of seed distribution and uniformity for sowing of solo crops, whereas uniformity was disturbed when seed mixtures were sowing in different sowing depths. Currently, there is still little or no information on comparing of mixture seeding performance of a seed drill under laboratory and field conditions. Therefore, the objective of this work is to assess the performance of the seed drill using pure barley, pure common vetch and mixtures of these crops in laboratory and field conditions. Optimization of the performance of seed drills for sowing seed mixtures of barley and vetch was also developed within the scope of this study.

## MATERIALS & METHODS

A combined seed drill with 18 rows and 12.5 cm of row spacing was employed for laboratory tests and field experiments as depicted in Fig. 1. The seed drill had separate metering systems for seeding and fertilizing. The seed metering unit has a studded roller while the fertilizer

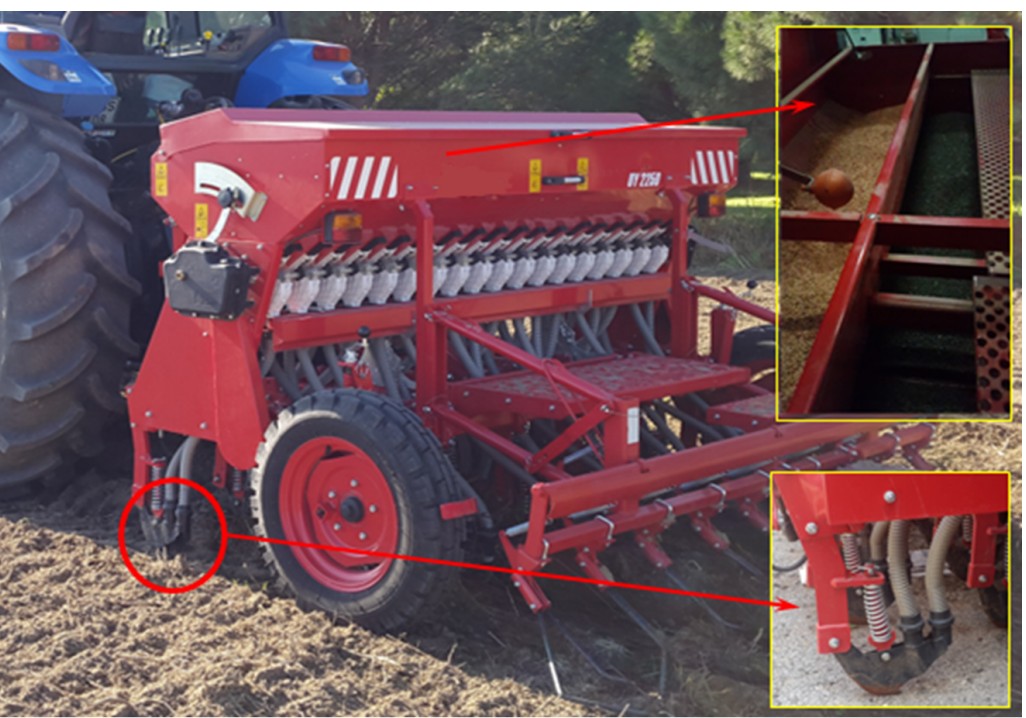

**Figure 1  Combined seed drill and mixed seeding condition.**

metering unit is equipped with a fluted roller. Seed and fertilizer coming from different hoppers were transferred to various rollers before they separately passed through different tubes. The seed and fertilizer released from the hoe-type furrow opener dropped into the soil. The mixed seeding condition was carried out using the barley in the seeding unit and the common vetch in the fertilizer unit as illustrated in Fig. 1.

The seeding and fertilizing rates were adjusted independently by a tri-cam mechanism that was placed for each metering system. Flow uniformity and distribution uniformity across rows were determined at varying seeding rates.

In the experiments, barley (*Hordeum vulgare* L.) with a thousand seed mass of 44.2 g and common vetch (*Vicia sativa* L.) with a thousand seed mass of 51.8 g were used.

The seed drill performance was investigated under the laboratory conditions in terms of seeding rate, flow uniformity and seed distribution uniformity across the rows by weighing tests. The metering unit was operated, seed flow was achieved and the flowing seeds were collected in the boxes during 30 s for each test. Then the collected seeds were weighed. Three replications were applied for each test.

Three different forward speeds (1.0, 1.5, and 2.0 m s$^{-1}$) and three different seeding rates (metering unit scale positions of 20, 60, and 100) were applied for determining flow evenness and seed distribution uniformity across the rows for the experiments. The scale positions of 20, 60 and 100 represent the seed rates of 90, 335 and 640 kg ha$^{-1}$ for vetch while the seed rates were 90, 320 and 635 kg ha$^{-1}$ for barley.

**Table 1  Qualitative evaluation traits corresponding to values of seed flow evenness and seed distribution uniformity across the rows.**

| Seed flow evenness, CV (%) | Seed distribution uniformity across the rows, CV (%) | Evaluation |
|---|---|---|
| <1 | <4 | Very good |
| 1–2 | 4–6.3 | Good |
| 2–3 | 6.3–8.9 | Moderate |
| 3–4 | 8.9–12.5 | Sufficient |
| >4 | >12.5 | Insufficient |

Seed weights obtained from each row unit were measured using a digital scale and the recorded data were evaluated as coefficient of variation (CV, %) values based on *Önal (2017)* are shown in Table 1.

The seed and fertilizer meters were driven by an electronically controlled gear motor to provide synchronization between the forward speed of the machine and the rotational speed of the metering unit.

The seed drill was tested to determine the in-row seed spacing uniformity using barley in the seeding unit and vetch in the fertilizing unit on a sticky belt test stand. To evaluate the seeding quality, sticky belt tests were conducted using pure barley (100%)100:0, pure common vetch (100%) 0:100 as well as different mixtures of barley:vetch (75:25, 50:50, 25:75) for seeding rates of 200 kg ha$^{-1}$ and 100 kg ha$^{-1}$ for barley and vetch, respectively.

The seeding quality of the seed drill was evaluated based on the variation factor ($V_f$) and the goodness criterion ($\lambda$) given by *Önal (2017)*. The numbers of seeds on 250 segments were determined and evaluated for each test. The experiments were conducted at forward speeds of 1.0, 1.5, and 2.0 m s$^{-1}$.

The seed drill was also tested to determine the in-row plant spacing uniformity under the field conditions. The field experiments were conducted at the forward speed of 1.5 m s$^{-1}$ using pure barley (100%), pure common vetch (100%) as well as different mixtures of barley:common vetch (75:25, 50:50, 25:75) for seeding rates of 200 kg ha$^{-1}$ and 100 kg ha$^{-1}$ for barley and vetch, respectively. The experiments were carried out as per randomized complete block design (RCBD) in a regular arrangement. The least significant difference (LSD) test was used for comparing of the means statically. The field tests were conducted under rainfed conditions and fertilizer was not applied during vegetation period. The soil profile at depths of 0–20 cm and 20–40 cm was characterized as silt-clay with a pH of 8.2 and clay-loamy with a pH of 7.8, respectively. The conventional soil tillage method was applied using plough, cultivator, disk harrow and roller. Each treatment was conducted in the 135 m$^2$ of area. The plant emergence measurements were done on 8 January 2018 while the harvesting was done on 18 April 2018. A view from the experimental area is shown in Fig. 2.

The in-row seed and plant spacing uniformity was evaluated by the values of variation factor ($V_f$) and the goodness criterion ($\lambda$) which defined by computer-aided classification as given. While the $V_f$ value describes how well the data fit a Poisson distribution, $\lambda$ value

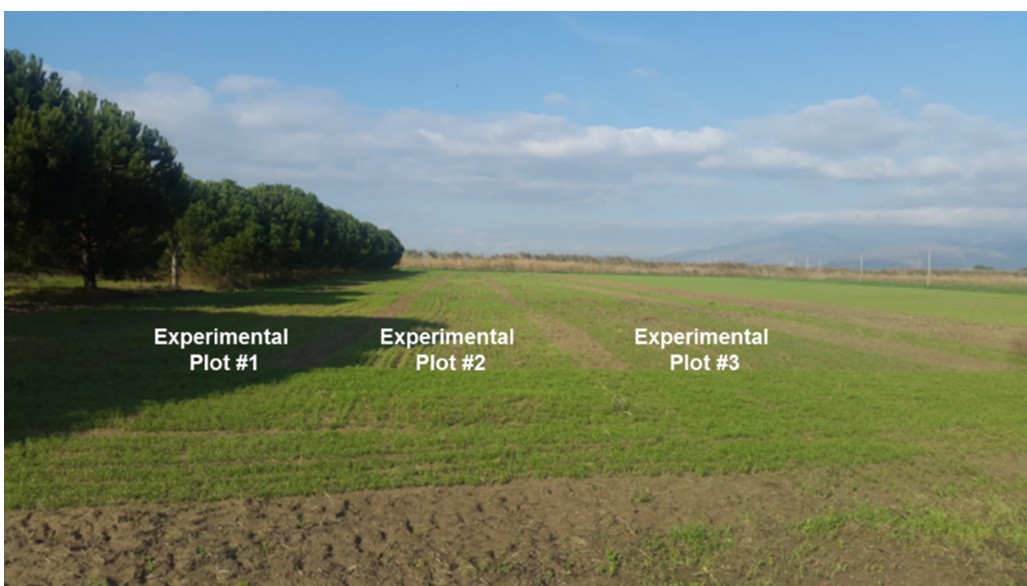

**Figure 2** The experimental area having sole crops (barley and vetch) and seeding mixtures.

defines the percentage of segments with 1, 2, and 3 seeds or plants. Data were collected and evaluated as previously described in *Turkusay & Yazgi (2024)*.

The calculations of variation factor ($V_f$) and variance ($S^2$) of the seed and plant distribution were given below as Eqs. (1) and (2) (*Griepentrog, 1994*).

$$V_f = \frac{S^2}{\mu} \tag{1}$$

$$S^2 = \frac{\sum_{i=1}^{n} x_i^2 f_i - \left(\left(x_i f_i\right)^2 / n\right)}{n-1}. \tag{2}$$

In Eqs. (1) and (2), $\mu$ is the average number of seeds per segment, $x_i$ is the expected number of seeds or plants in the segment, $f_i$ is the segment ratio that is the percentage of the segments with different numbers of seeds or plants, and $n$ is the total sample number.

The character of in-row seed/plant spacing was determined based on the "$V_f$" value obtained from the experiments. In this determination, if $V_f > 1.1$, it indicates there are undesired misses and multiples within the in-row seed/plant spacing. If $0.9 < V_f < 1.1$, this indicates the in-row seed/plant spacing matches a Poisson distribution. If $V_f < 0.9$, this indicates the in-row seed/plant spacing may be characterized as precision seeding (*Önal, 2017*).

The uniformity quality of the in-row seed/plant spacing was determined by a goodness criterion ($\lambda$) that defines the percentage segments with one, two, and three seeds. This criterion is interpreted using Table 2 (*Önal, 2017*).

In the goodness criterion evaluation, the average of number of seeds/plants per segment, $\mu$, was chosen to be $\mu = 2$, and the segment length, a, was calculated according to Eq. (3)

**Table 2  The quality of in-row seed/plant spacing uniformity.**

| Goodness criterion, λ (%) | Evaluation |
|---|---|
| ≥72 | Very good |
| 72–65 | Good |
| 65–55 | Moderate |
| <55 | Insufficient |

**Table 3  Treatments used in the weighing experiments.**

| Forward Speed (ms$^{-1}$) | Scale position | Replication number |
|---|---|---|
| | 20 | 3 |
| 1.0 | 60 | 3 |
| | 100 | 3 |
| | 20 | 3 |
| 1.5 | 60 | 3 |
| | 100 | 3 |
| | 20 | 3 |
| 2.0 | 60 | 3 |
| | 100 | 3 |

(*Önal, 2017*).

$$a = \frac{100\ \mu\sigma}{bN} \qquad\qquad (3)$$

where, $\sigma$ is thousand seed mass (g/1,000 seeds), $b$ is row spacing (cm), and $N$ is the seeding rate (kg ha$^{-1}$). The segment length in sticky belt tests conducted with mixed seed was calculated as $\mu = 2$ based on the seed that has the higher ratio in the seed mixture and the numbers of barley and common vetch seeds in the same segment were determined separately.

Based on this calculation, the segment length was calculated as 3.5 and 8.3 cm for pure barley and vetch, respectively.

The treatments used in the laboratory experiments, namely weighing and sticky belt experiments are tabulated in Tables 3 and 4, respectively, while treatments used in the field experiments were illustrated in Fig. 3.

## RESULTS AND DISCUSSION

The results were given in two parts included laboratory and field experiments. Portions of this text were previously published as part of a conference paper (*Yazgi et al., 2018*).

### Flow tests

Based on the findings, the average seeding rate values were determined that varied between zero and 632 kg ha$^{-1}$ for barley in the seeding unit, while these values were between zero

**Table 4  Treatments used in the sticky belt experiments.**

| Forward Speed (ms$^{-1}$) | Mixture ratio (barley:vetch) | Replication number |
|---|---|---|
| | 100:0 | 3 |
| | 75:25 | 3 |
| 1.0 | 50:50 | 3 |
| | 25:75 | 3 |
| | 0:100 | 3 |
| | 100:0 | 3 |
| | 75:25 | 3 |
| 1.5 | 50:50 | 3 |
| | 25:75 | 3 |
| | 0:100 | 3 |
| | 100:0 | 3 |
| | 75:25 | 3 |
| 2.0 | 50:50 | 3 |
| | 25:75 | 3 |
| | 0:100 | 3 |

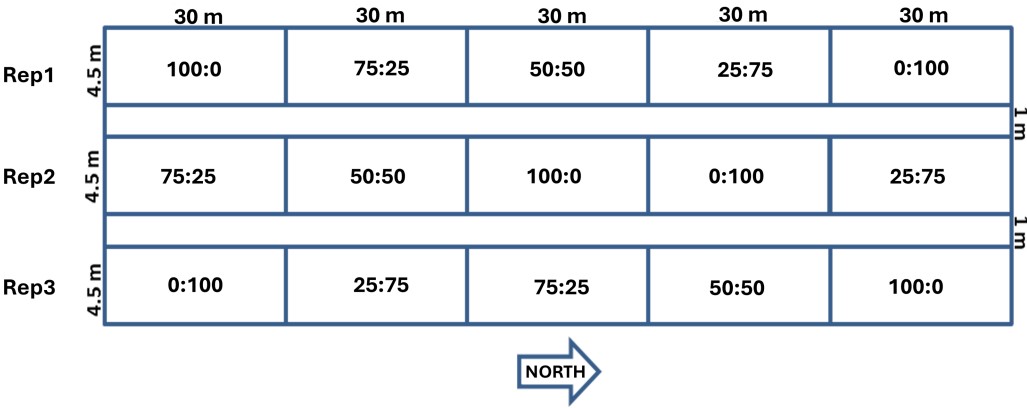

**Figure 3  Treatments used in the field experiments.**

and 648 kg ha$^{-1}$ for the vetch in fertilizing unit. Weighing measurements for barley in the seeding unit and vetch in the fertilizing unit are tabulated in Tables 5 and 6, respectively.

Based on the results of the flow tests with barley, the performance of the seeder was found sufficient for flow evenness and seed distribution uniformity across the rows. The seeder also had a very good quality in vetch seeding at different forward speeds and seed rates for flow evenness and seed distribution uniformity across the rows. As a result, the seed drill had a satisfactory quality under the laboratory conditions in terms of flow evenness and in-row seed spacing uniformity of row units for both barley and vetch seeds.

**Table 5  Weighing measurements for barley in the seeding unit (*Yazgi et al., 2018*).**

| Forward speed (ms$^{-1}$) | Scale position | Seeding rate (kg ha$^{-1}$) | Seed flow evenness | | Seed distribution uniformity across the rows | |
|---|---|---|---|---|---|---|
| | | | CV (%) | Evaluation | CV (%) | Evaluation |
| | 20 | 84 | 3.2 | Sufficient | 8.8 | Moderate |
| 1.0 | 60 | 319 | 2.3 | Moderate | 7.0 | Moderate |
| | 100 | 626 | 5.5 | Insufficient | 4.2 | Good |
| | 20 | 90 | 2.9 | Moderate | 6.9 | Moderate |
| 1.5 | 60 | 349 | 1.0 | Good | 9.8 | Sufficient |
| | 100 | 642 | 2.4 | Moderate | 10.7 | Sufficient |
| | 20 | 92 | 2.4 | Moderate | 7.2 | Moderate |
| 2.0 | 60 | 341 | 2.5 | Moderate | 7.8 | Moderate |
| | 100 | 655 | 1.4 | Good | 8.6 | Moderate |

**Table 6  Weighing measurements for vetch in the fertilizing unit (*Yazgi et al., 2018*).**

| Forward Speed (ms$^{-1}$) | Scale position | Seeding rate (kg ha$^{-1}$) | Seed flow evenness | | Seed distribution uniformity across the rows | |
|---|---|---|---|---|---|---|
| | | | CV (%) | Evaluation | CV (%) | Evaluation |
| | 20 | 92 | 2.1 | Moderate | 1.4 | Very good |
| 1.0 | 60 | 326 | 0.3 | Very good | 0.5 | Very good |
| | 100 | 650 | 0.5 | Very good | 0.7 | Very good |
| | 20 | 90 | 1.2 | Good | 0.5 | Very good |
| 1.5 | 60 | 320 | 0.3 | Very good | 0.4 | Very good |
| | 100 | 631 | 1.2 | Good | 0.6 | Very good |
| | 20 | 90 | 1.2 | Good | 0.9 | Very good |
| 2.0 | 60 | 317 | 0.9 | Very good | 0.8 | Very good |
| | 100 | 624 | 0.9 | Very good | 0.7 | Very good |

## Sticky belt tests

The results of sticky belt tests conducted at five different barley:vetch mixture ratios (100:0, 75:25, 50:50, 25:75, and 0:100) and three different forward speeds (1.0, 1.5, and 2.0 ms$^{-1}$) are tabulated in Table 7 related to seeding characterization and quality.

Based on the sticky belt results, the seed drillcould sow barley and vetch mixtures in certain ratios with the in-row seed spacing uniformity characterized as random or precision seeding, generally.

The $V_f$ values decreased with increasing forward speed and the seeding character got better. The 75:25 and 0:100 ratios have irregular trend (Table 7). Additionally, the seed distribution quality of the seed mixture was found in "very good" quality, generally. Most of the goodness criteria values were found higher than 72%. This means that the combined seed drill was capable of sowing barley and vetch seeds and their mixtures, with ratios of satisfactory quality (Table 7).

**Table 7  Seed distribution uniformity results at different mixture ratios (*Yazgi et al., 2018*).**

| Mixture ratio (Barley:Vetch) | Forward speed (ms$^{-1}$) | $V_f$ | Seeding characterization | λ(%) | Seeding quality |
|---|---|---|---|---|---|
| | 1.0 | 0.98 | Random seeding | 71.9 | Good |
| 100:0 | 1.5 | 0.97 | Random seeding | 75.2 | Very good |
| | 2.0 | 0.87 | Precision seeding | 75.2 | Very good |
| | 1.0 | 0.78 | Precision seeding | 79.3 | Very good |
| 75:25 | 1.5 | 0.82 | Precision seeding | 78.8 | Very good |
| | 2.0 | 0.70 | Precision seeding | 83.6 | Very good |
| | 1.0 | 1.20 | Undesirable seeding | 69.2 | Good |
| 50:50 | 1.5 | 0.98 | Random seeding | 73.2 | Very good |
| | 2.0 | 0.85 | Precision seeding | 79.2 | Very good |
| | 1.0 | 0.49 | Precision seeding | 83.2 | Very good |
| 25:75 | 1.5 | 0.46 | Precision seeding | 86.0 | Very good |
| | 2.0 | 0.43 | Precision seeding | 83.6 | Very good |
| | 1.0 | 1.05 | Random seeding | 72.8 | Very good |
| 0:100 | 1.5 | 1.18 | Random seeding | 70.4 | Good |
| | 2.0 | 1.28 | Undesirable seeding | 66.4 | Good |

**Table 8  Plant distribution uniformity results at different mixture ratios at forward speed of 1.5 m s$^{-1}$.**

| Mixture ratio (Barley:Vetch) | $V_f$ | Seeding characterization | λ (%) | Seeding quality |
|---|---|---|---|---|
| 100:0 | 0.45 | Precision seeding | 80.8 | Very good |
| 75:25 | 0.54 | Precision seeding | 82.8 | Very good |
| 50:50 | 0.60 | Precision seeding | 78.4 | Very good |
| 25.75 | 0.36 | Precision seeding | 78.0 | Very good |
| 0:100 | 0.77 | Precision seeding | 86.0 | Very good |

## Field experiments

The results from mixed seeding experiments with five different barley:vetch mixture ratios (100:0, 75:25, 50:50, 25:75, and 0:100) at a forward speed of 1.5 m s$^{-1}$ are given in Table 8 related to seeding characterization and quality under the field conditions.

Based on the results of field tests, the plant distribution quality of the mixture was found to be "very good" and all criteria values remained higher than 72% indicating the seeding characterization as precision seeding under the field conditions as displayed in Table 8. The examples of in-row plant distribution of the barley and vetch mixtures at different ratios are given in Fig. 4.

## Comparison of sticky belt and field performance of the seed drill

The comparative results obtained from mixed seeding experiments in the laboratory and the field conditions at a forward speed of 1.5 m s$^{-1}$ have been presented in Fig. 5. The results exhibited that the sticky belt and field test findings for seed/plant spacing uniformity were in harmony with each other. Additionally, the performance of the seeder in the field was found higher than those determined in the laboratory for pure barley and vetch in goodness

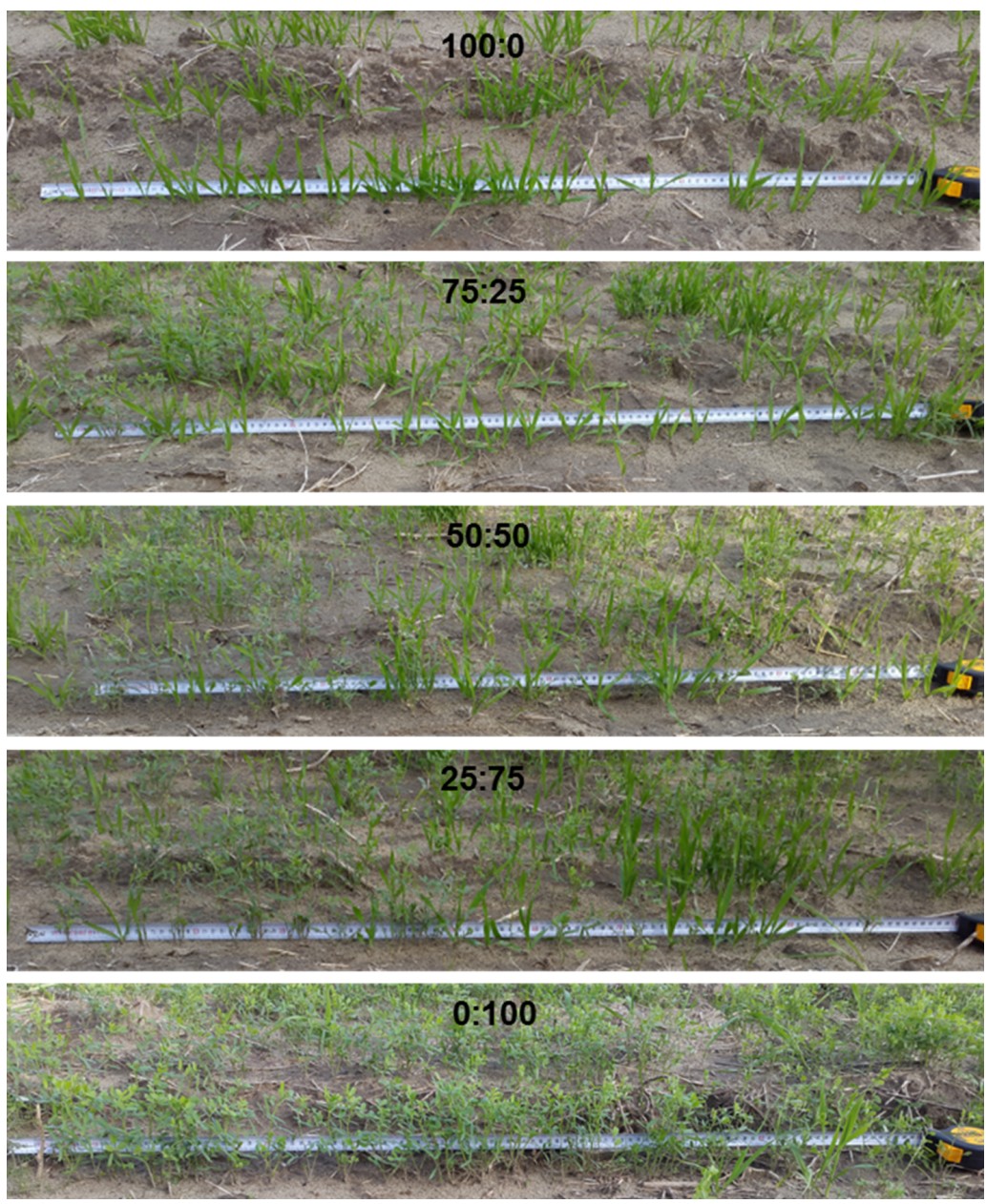

**Figure 4** In-row seed distribution of the barley and vetch mixtures at different ratios.

criterion and variation factor. The mixture of 75:25 had a better plant distribution in the field than that of the seed distribution in the laboratory, while the seed distribution also demonstrated higher quality than that of the plant distribution recorded for the mixture of 25:75.
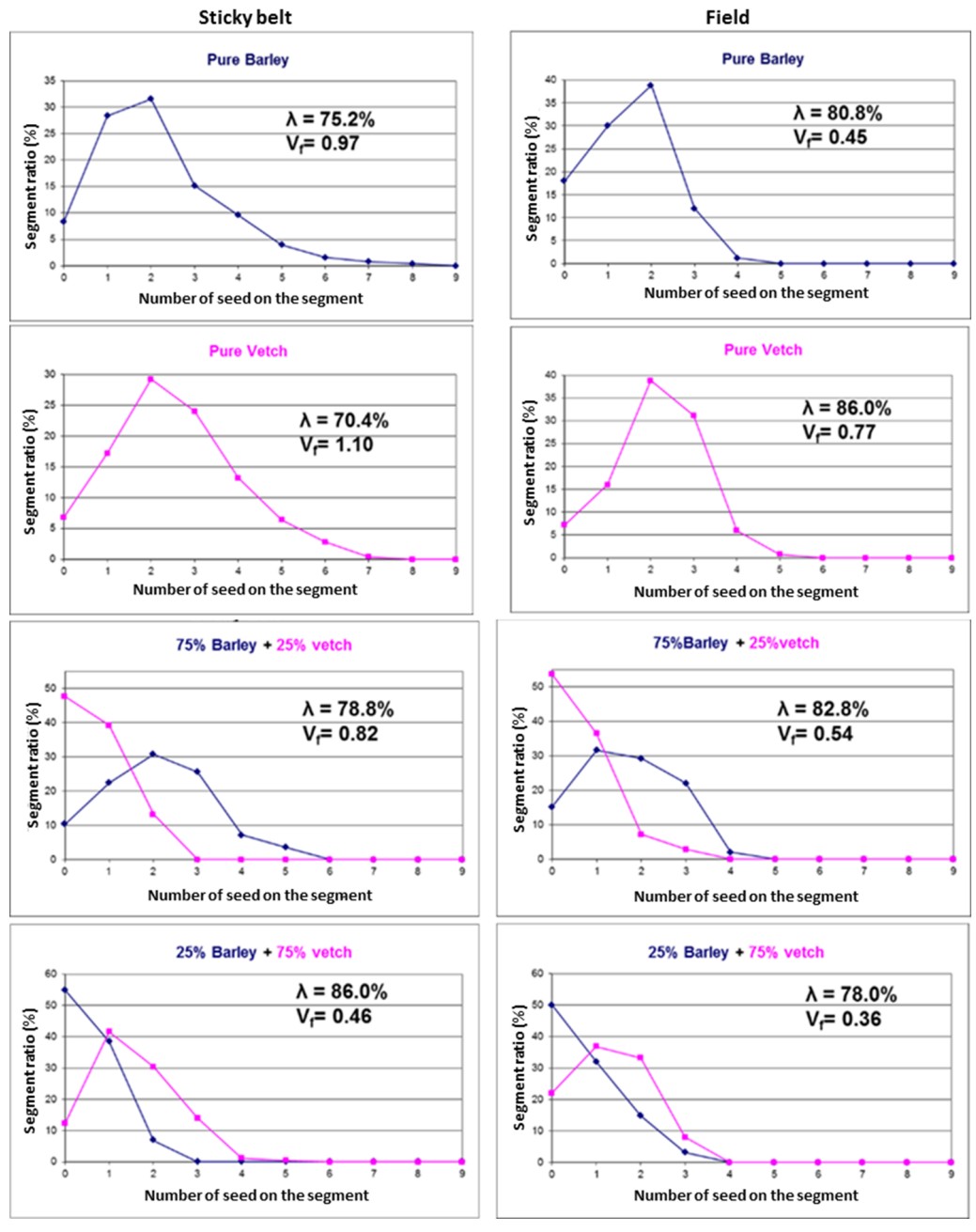

**Figure 5  The sticky belt and field results for barley and vetch mixtures at different ratios.**

## Plant emergence, yield and quality evaluation

After harvesting, the data were analyzed for the yield and feed quality. Plant height (cm), fresh biomass yield (kg ha$^{-1}$), dry matter content (%) and dry matter yield (kg ha$^{-1}$) values were determined and the results were subjected to statistical analysis as shown in Table 9.

As per recorded findings, the highest main stem length value was obtained as 124.3 cm with the ratio of 75:25, while the lowest main stem length was obtained as 74.0 cm for

**Table 9  Plant height, fresh biomass yield, dry matter content and dry matter yield of different seeding mixtures of barley and vetch.**

|  | Mixture ratio of barley:vetch | | | | | Average | LSD (5%) |
|---|---|---|---|---|---|---|---|
|  | 100:0 | 75:25 | 50:50 | 25:75 | 0:100 |  |  |
| Plant height (cm) | 100.2[c] | 124.3[a] | 122.2[ab] | 118.7[b] | 74.0[d] | 107.9 | 3.5 |
| Fresh biomass yield (kg ha$^{-1}$) | 29,100[c] | 40,850[b] | 43,030[a] | 37,770[bc] | 42,620[ab] | 38,670 | 2,030 |
| Dry matter content (%) | 23.03[a] | 21.34[c] | 22.19[b] | 22.01[b] | 16.93[d] | 21.10 | 0.72 |
| Dry matter yield (kg ha$^{-1}$) | 6,700[d] | 8,720[b] | 9,550[a] | 8,310[bc] | 7,210[c] | 8,100 | 400 |

Notes.
  Means with the same letter in the same row are not significantly different from each other for each table (95% significance level).

the ratio of 0:100 condition. Additionally, the yield and feed quality results showed that the highest fresh biomass yield was 43,030 kg ha$^{-1}$ for the mixture ratio of 50:50 while the lowest corresponding value was 29,100 kg ha$^{-1}$ recorded for the mixture ratio of 100:0. Moreover, the highest dry matter content (23.03%) was found for the mixture ratio of 100:0 and the lowest value was 16.93% for 0:100 mixture ratio. The highest dry matter yield was obtained for the mixture ratio of 50:50 with a value of 9,550 kg ha$^{-1}$, while the lowest was obtained for the mixture ratio of 100:0 with the value of 6,700 kg ha$^{-1}$ (Table 9). The results obtained from the experiments were found to be compatible with *Soya et al. (1999)*. They inferred that intercropping of cereal and legumes remained effective by recording significantly higher biomass yield on per unit land area basis. It was also inferred that spatial–temporal optimization of component crops played a crucial role in boosting the overall productivity of the intercropping systems. However, *Iqbal, Iqbal & Abbas (2018)* also reported that cereals tend to dominate the legumes in an intercropping system which resulted in the significant loss of yield of legume compared to sole crop.

## CONCLUSIONS

Based on the findings in this work, it can be stated that the combined seed drill could be effectively used for the seeding of barley and vetch mixtures at different mixture ratios as per response variables determined in the laboratory and field conditions.

Since separate setting of both seeding and fertilizer units were precisely determined desired mixture ratios were accurately achieved. Consequently, the overall seeding and seed mixing quality directly influenced and enhanced.

The comparison of laboratory and field results revealed that the seeder conductivity remained quite good and the seeder had similar results in these conditions. Also, the performance of the seeder in the field was found to be higher than the laboratory's for some operational parameters. These findings showed that the performance of soil engagement components was as higher importance as performance of the metering unit.

Consequently, the seed mixtures were dropped in the soil by metering unit and soil components were placed and covered the seeds, precisely resulting in a greater plant emergence so that the overall yield was significantly increased.

### Funding
This work was granted by the Ege University Scientific Research Projects Coordination Unit. Project Number 2015ZRF-024. The funders had no role in study design, data collection and analysis, decision to publish, or preparation of the manuscript.

### Grant Disclosures
The following grant information was disclosed by the authors:
The Ege University Scientific Research Projects Coordination Unit: 2015ZRF-024.

### Competing Interests
The authors declare there are no competing interests.

### Author Contributions
- Arzu Yazgi conceived and designed the experiments, performed the experiments, analyzed the data, prepared figures and/or tables, authored or reviewed drafts of the article, and approved the final draft.
- Tuncay Gunhan conceived and designed the experiments, performed the experiments, analyzed the data, authored or reviewed drafts of the article, and approved the final draft.
- Behcet Kir conceived and designed the experiments, performed the experiments, analyzed the data, authored or reviewed drafts of the article, and approved the final draft.
- Gulcan Demiroglu Topcu analyzed the data, authored or reviewed drafts of the article, and approved the final draft.
- Erdem Aykas analyzed the data, authored or reviewed drafts of the article, and approved the final draft.

### Data Availability
The raw measurements are available in the Supplemental Files.

### Supplemental Information
Supplemental information for this article can be found online at http://dx.doi.org/10.7717/peerj.19014#supplemental-information.

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
