# Peer review of "Evaluation of a seed drill for barley and vetch sowing as a function of varying seeding ratios"

_PeerJ, doi:10.7717/peerj.19014_

## Round 0.1 · original submission · Major Revisions

The current MS has been reviewed by 3 independent reviewers. Although I didn't like to comment on language editing, but all three reviewers highlighted on language editing. After personally reading the MS, I agree with reviewers' assessment that it requires language editing for better readability.

-Fig 1, and Fig 2 should be listed as supplementary figure.

-Add description of technical or biological replicates in figures/tables.

-Please include "Statistical analysis" section in the Materials and methods to explain the statistical methods used in the experiment.

Please incorporate all the changes as requested by reviewers.

Reviewer 1 ·

Basic reporting

As I made some comments on English some minor revision is required.
Figures and Tables do not have any problems
Satisfactory number of references was cited in corresponding parts of the manuscript.

Experimental design

.

Validity of the findings

.

Additional comments

I reviewed article entitled" Evaluation of a seed drill............................." by Yazgi et al.
This is straightforward experimental study having its objectives are clearly stated and methods were carried out related to the objectives of the work. Definition of the problem has been clearly presented in the manuscript. I have no problem any technical and ethical standard of the investigation.
I have some editorial corrections and suggestions as I tried to list below, once the manuscript is revised based on them I guess it will be ready for publication.
Line 75 .......the performance
Line 81 ......of this work was to.....
Line 103-105............the rows for the experiments.
Line 137 In Equations 1 and 2 ,
Line 159 Flow tests use such structure rather than "experiments" . Revise this for the subtitles and anywhere in the text. Sticky best test etc
Line 211 results of what ???? not complete
Line 229-230 not very clear it hink you want to say below :
Since separate setting of both seeding and fertilizer units were precisely determined desired mixture ratios were accurately achieved. Consequently the overall seeding and mixing quality (quality of what ???) directly influenced and enhanced.

Results of the study were clearly presented without having any flaws, and related to those determined in past works

Reviewer 2 ·

Basic reporting

The primary reporting is straightforward. The references are adequate and give sufficient context. The layout is good, and all the necessary raw data, figure graphs, etc., were available. The findings answer the hypothesis.

The English, however, needs serious work. I stopped editing language and sentence construction after two or three paragraphs. I suggest the authors hire someone to do a proper English edit before publishing this manuscript. Some of the wording feels like a direct translation via an app, which is confusing some statements.

Experimental design

The experimental design is sound and falls within the journal's scope. The research question was well-defined. The authors did well in describing the methods they employed, making it easy to duplicate.

Validity of the findings

The results are valid for this experiment. Testing two or more varieties of each crop would have been more relevant because varieties are not all similar. If they had used three of four different types of each, the work might have had even more impact—the same with the seeder used.

They draw the right conclusions.

Additional comments

The authors fail to mention whether the field experiments were conducted under rainfed or irrigated conditions. It would give more insight into the biomass values they obtained. They also did not mention fertilisation of the field crops.

Annotated reviews are not available for download in order to protect the identity of reviewers who chose to remain anonymous.

Reviewer 3 ·

Basic reporting

no comment

Experimental design

no comment

Validity of the findings

no comment

Additional comments

Dear Authors,
Try to follow the comments and suggestions, which are remarked in the word file to improve the manuscript.

Report on PeerJ-ID-107226
Dear PeerJ Team,
The manuscript " Evaluation of a seed drill for barley and vetch sowing as a function of varying seeding ratios " aims to evaluate the performance of a seed drill for seeding barley and vetch mixtures at 100:0, 0:100, 75:25, 50:50, 25:75 ratios and determine their impact on Plant height, Fresh biomass yield, Dry matter content, and Dry matter yield.

I have read the manuscript and supplementary file carefully. The manuscript is written in a good style, and I appreciate the work and effort of the authors.

However, I found a few issues, which are to be addressed.

Line 27 " were found as " → were found to be
Line 30 " ranged within " → ranged between
Line 31 " values had range " → values had a range
Line 37 " as function of " → as a function of
Line 39 " nutritional quality " → Please clarify the relationship of the keyword to the content of the article
Line 42 " namely cultivation of " → namely the cultivation of
Line 43 " on the same of land " → on the same land
Line 44 " on per unit land area " → on a per unit land area
Line 46 " temporal dimension " → temporal dimensions
Line 50 " however it is " → however, it is
Line 54 " for weed controlling " → for weed control
Line 57 " seeds at right depth " → seeds at the right depth
Line 58 " factors which ultimately contribute towards germination" → factors that ultimately contribute to the germination
Line 59 " centrifugal fertilizer " → centrifugal fertilizer spreaders ??
Line 61 " however this " → however, this
Line 63 " agronomical " → Is the term “agronomic” related to the content of the paragraph?
Line 63 " previous investigation " → previous investigations
Line 65 " inferred that comparatively " → inferred that a comparatively
Line 65 " lesser ratio of " → Please add the ratios mentioned in the reference.
Line 69 " obtaining higher herbage " → obtaining a higher herbage
Line 71 " parameter on seed distribution " → parameter of seed distribution
Line 77 " In a study conducted by Yazgi et al. (2017) and it was " → In a study conducted by Yazgi et al. (2017) it was
Line 77 " performance determined " → performance was determined
Line 81 " this work to assess " → this work is to assess
Line 83 " of seed drill for sowing " → of seed drills for sowing
Line 86 " " → Please add a table showing the initial soil properties, texture, and available nutrients.
Please clarify the method of preparing the soil before seeding (type of tillage) and the amount of fertilizer added to the field.
It is preferable to add a table showing the treatments used in the laboratory experiment and the treatments used in the field experiment and their details.
Line 87 " with 18 rows, and " → with 18 rows and
Line 92 " the hoe type " → the hoe-type
Line 95 " that placed for each " → that was placed for each
Line 102 " seeds were collected for 30 s " → Please explain the highlighted text.
Line 104 " metering unit scale positions of 20, 60, and 100 " → Please add the average of the corresponding seed rate for each scale position.
Line 104 " determining of flow evenness " → determining flow evenness
Line 114 " pure barley (100%), pure common vetch (100%) " → pure barley (100%) 100:0, pure common vetch (100%) 0:100
Line 119 " The experiment were conducted " → The experiment was conducted
Line 126 " " → Please add the test used to compare the means.
Line 126 " (RCBD) in regular arrangement " → (RCBD) in a regular arrangement
Line 131 " plant distribution were given " → plant distribution is given
Line 137 " average of number of seeds " → average number of seeds
Line 139 " different number " → different numbers
Line 154 " on the seed which has " → on the seed that has
Line 164 " tabulated in Table " → tabulated in Tables
Line 164 " " → It appears that the data in Tables 3 and 4 belong to a previous study as indicated. Please explain its relationship to the current experiment in terms of the similarity of the conditions of the two experiments.
Line 175 " " → It appears that the data in Table 5 are from a previous study as indicated. Please explain its relationship to the current experiment in terms of the similarity of the conditions of the two experiments.
Line 176 " sow a barley and vetch mixtures " → sow barley and vetch mixtures
Line 179 " with increasing of forward " → with increasing forward
Line 179 " except pure vetch (0:100) " → The 75:25 ratio has an irregular trend.
Line 183 " with the ratios of satisfactory " → with ratios of satisfactory
Line 187 " at forward speed of " → at a forward speed of
Line 188 " seeding characterization and quality under the field conditions " →
Line 189 " seeding characterization and quality under the field conditions " → Please rephrase the sentence, it was mentioned previously.
Line 197 " conditions at forward speed " → conditions at a forward speed
Line 199 " those of determined in the laboratory for " → those determined in the laboratory for
Line 205 " Plant emergence, yield and quality evaluation " → Please rephrase the title to match the title of Table 7.
Line 205 " " → Please add the area from which the sample was taken, the method of measurement for each trait, and the harvest date (specific date or at maturity).
Line 209 " with the ratio of 75:25 " → with a ratio of 75:25
Line 211 " green grass " → fresh biomass
Line 217 " experiments was found " → experiments were found
Line 219 " it was also inferred that spatial temporal optimization of component crops played a crucial role in boosting the overall productivity of the intercropping systems " → Please clarify the text.
Line 219 " spatial temporal " → spatial-temporal
Line 222 " in intercropping system " → in an intercropping system
Line 229 " Because of the precision seeding rate setting " → Please clarify the text.
Line 229 " seeding and fertilizing unit " → seeding and fertilizing units
Line 235 " that performance of soil engagement components was as higher as performance of metering unit " → Please clarify the text.
Line 237 " soil components placed " → soil components were placed


Line 3 " Qualitative evaluation adjectives " → Qualitative evaluation traits
Line 2 " Table 6. Plant distribution uniformity results at different mixture ratios. " → Table 6. Plant distribution uniformity results at different mixture ratios at forward speed of 1.5 m s-1.
Line 2 " Table 7/ Plant height, fresh biomass yield, dry matter content and…. " → Please add the significance level to column LSD (5%).

Annotated reviews are not available for download in order to protect the identity of reviewers who chose to remain anonymous.

---

## Round 0.2 · accepted · Accept

The authors did address all comments and suggestions as suggested before in this revised version. In this revised form, this MS is good to go for publication.

Reviewer 1 ·

Basic reporting

Based on my scanning and reviewing of revised manuscript by Yazgi et al. It appears that manuscript has been revised based on the questions, comments and suggestions of the reviewers. Therefore if this work is published as its revised format I have no problem.
Regards,

Experimental design

Please see section 1

Validity of the findings

Please see section 1

Additional comments

None

Reviewer 3 ·

Basic reporting

No comment

Experimental design

No comment

Validity of the findings

No comment

Additional comments

Report on peerj-reviewing-107226-v1

Dear PeerJ Team,
I would like to thank you for the effort and expertise that you contributed to reviewing the manuscript " Evaluation of a seed drill for barley and vetch sowing as a function of varying seeding ratios ".
I would like to thank the authors for their time and effort to revise the manuscript.
I have read all the replies by the authors, and the issues have been addressed carefully.

Sincerely,

Annotated reviews are not available for download in order to protect the identity of reviewers who chose to remain anonymous.